# Retrospective Analysis of Clinical Characteristics and Disease Outcomes in Children and Adolescents Hospitalized Due to COVID-19 Infection in Tunisia

**DOI:** 10.3390/v16050779

**Published:** 2024-05-14

**Authors:** Aida Borgi, Khaoula Meftah, Ines Trabelsi, Moe H. Kyaw, Hela Zaghden, Aida Bouafsoun, Fatma Mezghani, Nada Missaoui, Alya Abdel Ali, Leila Essaddam, Haifa Khemiri, Sondes Haddad-Boubaker, Khedija Boussetta, Monia Khemiri, Saida Ben Becher, Samir Boukthir, Henda Triki, Khaled Menif, Hanen Smaoui

**Affiliations:** 1Pediatric Intensive Care Unit, Children’s Hospital of Tunis, Tunis 1007, Tunisia; aidabdoc@yahoo.fr (A.B.); menifk@yahoo.fr (K.M.); 2Laboratory of Microbiology, Children’s Hospital of Tunis, Tunis 1007, Tunisia; meftahkhaoula@gmail.com (K.M.);; 3Pediatric Department B, Children’s Hospital of Tunis, Tunis 1007, Tunisia; drinestrabelsi@gmail.com (I.T.);; 4Pfizer Inc., New York, NY 10017, USA; 5Pfizer Inc., Tunis 1007, Tunisia; hela.zaghden@pfizer.com; 6Pediatric Department A, Children’s Hospital of Tunis, Tunis 1007, Tunisia; 7Pediatric Department C, Children’s Hospital of Tunis, Tunis 1007, Tunisia; 8Department of Pediatrics and Emergency, Children’s Hospital of Tunis, Tunis 1007, Tunisia; 9Laboratory of Clinical Virology, WHO Regional Reference Laboratory for Poliomyelitis and Measles for the EMR, Institute Pasteur of Tunis, Tunis 1002, Tunisia

**Keywords:** observational study, COVID-19, pediatric population, clinical characteristics, disease severity, comorbidities, study outcomes

## Abstract

Due to low susceptibility of coronavirus disease of 2019 (COVID-19) in children, limited studies are available regarding COVID-19 in the pediatric population in Tunisia. The current study evaluated the incidence, clinical characteristics, and outcomes of severe acute respiratory syndrome coronavirus type 2 (SARS-CoV-2) infection among children hospitalized at Béchir Hamza Children’s Hospital. A retrospective cohort analysis was conducted using the hospital database between March 2020 and February 2022 with children aged ≤15 years with SARS-CoV-2 infection (confirmed by RT-PCR). A total of 327 COVID-19 hospitalized patients with a mean age of 3.3 years were included; the majority were male. Neurological disease (20%) was the most common comorbidity, while fever (95.3%) followed by cough (43.7%) and dyspnea (39.6%) were the most frequent symptoms reported. Severe disease with oxygen requirement occurred in 30% of the patients; 13% were admitted in the Intensive Care Unit. The overall incidence rate of COVID-19 hospitalization (in Tunis governorates) was 77.02 per 100,000 while the inpatient case fatality rate was 5% in the study population. The most prevalent circulating variant during our study period was Delta (48.8%), followed by Omicron (26%). More than 45% of the study population were <6 months and one-fourth (*n* = 25, 26.5%) had at least one comorbidity. Thus, the study findings highlight the high disease burden of COVID-19 in infants.

## 1. Introduction

Coronavirus disease 2019 (COVID-19), an infectious disease, has a substantial disease burden globally. The causative agent is the severe acute respiratory syndrome coronavirus type 2 (SARS-CoV-2). After the confirmation of the first case in December 2019, the disease witnessed exponential growth, with more than 768 million reported cases worldwide, culminating in a pandemic. Mortality data confirm 7 million deaths as of 2 August 2023 [1]. In early March 2020, Tunisia reported its first cases of COVID-19 [2]. Until 26 October 2023, Tunisia recorded 1,156,613 confirmed cases of SARS-CoV-2 infection with 29,494 deaths [3,4].

In the initial phase of the pandemic, COVID-19 infection was comparatively lower in children and adolescents than in the adult population in terms of both severity and frequency [5,6]. The reported cases of infections in children increased dramatically during the Omicron variant stage in early 2022, when there was relaxation in public health and social measures [7]. As the total global data on the incidence and mortality of pediatric COVID-19 cases are limited, it is challenging to assess the course and the effect of the disease in children [8].

Children play a pivotal role in COVID-19 transmission. Being asymptomatic, children may be responsible for the continuous spread of infection. However, a meta-analysis conducted by Chen et al. revealed that children are not actively involved in household transmission. The emergence of novel variants over time is the primary cause of increased transmissibility [9]. According to the study by Dong et al., about 51% of children developed relatively mild disease, while 4.4% were asymptomatic, and 5.3% had severe disease, with 0.6% having critical illness [10].

The introduction of vaccinations to stop the spread of COVID-19 proved successful in reducing disease outbreaks. Even though mRNA and inactivated virus vaccinations have been approved for pediatric patients, only mRNA vaccines are recommended for children in the majority of the African countries. According to the various published reports, the vaccination coverage for children was negligible in Africa [11]. Furthermore, the WHO Strategic Advisory Group of Experts (SAGE) reached a consensus regarding the primary series for COVID-19 vaccinations. They recommended it for healthy children, depending on the disease burden and economic impact within the specified age range as well as the opportunity cost for a particular country. The SAGE group has recommended integration of COVID-19 vaccination into routine immunization schedules (annual uptake) for eligible populations [12,13]. Children with underlying risk conditions such as chronic pulmonary disease, cardiovascular disease, and immunosuppression, and who are less than 1 year of age are at increased risk of severe disease and hospitalization. Therefore, it is important for family members of such children to get a COVID-19 vaccination to prevent occurrence of disease as well as mitigate transmission of disease to this most vulnerable group of children [14].

Given the seemingly low susceptibility of children to COVID-19, studies on the disease incidence, clinical characteristics, and outcomes in the pediatric population are limited in African countries, including Tunisia. Therefore, there is a dearth of data on the impact of COVID-19 in children and infants and the subset of patients hospitalized due to infection. Earlier, Borgi et al. conducted a retrospective case series study in Tunisia to evaluate the clinical characteristics and outcomes of COVID-19 in the pediatric population. The patients admitted to the intensive care unit (ICU) during the period when the Delta variant was predominant were analyzed. However, this study had the limitation of a smaller sample size [15]. Furthermore, for the 17 months following the first case detection in March 2020, high variability in the SARS-CoV-2 lineage was reported in Tunisia [16].

Thus, the current study was conducted to assess the incidence rate of hospitalization per 100,000 pediatric patients with SARS-CoV-2 infection in the age range of 0–15 years hospitalized at Béchir Hamza Children’s Hospital. The secondary objectives of the study were to describe the cases and proportion of underlying medical conditions, clinical presentations and outcomes, and to determine the inpatient case fatality rate of SARS-CoV-2 infection in the study population. The study also determined the viral load based on cycle threshold (CT) values (only for a subset of COVID-19 cases) and their relation to clinical outcomes.

## 2. Materials and Methods

### 2.1. Study Design and Data Source

This was a retrospective, observational database study of hospitalized COVID-19 positive children aged 0–15 years in Tunis between March 2020 and February 2022. The hospital’s electronic medical records system was utilized to identify the study population. Each study sample was assigned a unique laboratory code, and all personal and clinical information used in the study was handled with the utmost confidentiality.

The variables collected on the data entry form include the demographic data (age, sex, residence), medical history (underlying medical conditions), clinical presentation (signs and symptoms, duration of hospitalization, oxygen requirement, ICU admission, and disease outcome) and laboratory findings (PCR result with CT value viral load).

### 2.2. Study Population

The study database contained a total of 2807 hospitalized patients with both negative and positive samples. However, only positive samples with complete data were analyzed. The study included all children aged 0–15 years who were hospitalized at Béchir Hamza Children’s Hospital in the governorate of Tunis between March 2020 and February 2022 and had tested positive for SARS-CoV-2 infection using RT-PCR at the microbiology laboratory, with the results being recorded in the database. The study excluded patients aged > 15 years and patients without a laboratory confirmed SARS-CoV2 infection. The study also analyzed patients admitted to ICU by age group and comorbidity.

Béchir Hamza Children’s Hospital, a first-line center, primarily caters to the medical needs of an urban settlement of more than 2 million habitants. It is also a referral center for all the governorates in Tunisia, offering wider population coverage. Therefore, almost 100% of children’s COVID-19 hospitalizations in Tunis occurred at Béchir Hamza Children’s Hospital.

### 2.3. Baseline Variables and Study Outcomes

The baseline variables analyzed during the study included demographic data and underlying medical conditions. The study outcomes included descriptions of clinical presentation; assessment of severity of COVID-19 in hospitalized children (oxygen requirement, ICU admission, need for assisted ventilation, and disease outcome); CT value viral load; descriptions of deceased population with respect to age and underlying comorbidities and assessing the incidence rate of hospitalization and inpatient case fatality rate of COVID-19 among the hospitalized children aged < 15 years.

### 2.4. Complete Genome Sequencing and Genome Analysis

The full genome sequencing, sequences analysis and variant assignment were achieved, as previously described, at Pasteur Institute of Tunis and collaborative centers [17].

### 2.5. Statistical Analysis

A non-probabilistic sampling strategy was adopted, and all eligible cases were included in the analysis. The sample size was not calculated mathematically, as the entire population was exhaustively included. Data analysis was conducted using R v4.3.1 software. The data were summarized using descriptive statistics. Categorical variables were provided in terms of frequencies and percentages, while continuous variables were shown as means with standard deviations or as medians with interquartile ranges, as appropriate.

The incidence rate of hospitalization per 100,000 affected patients was calculated using the number of positive cases during the study period divided by the overall pediatric population aged 0–15 years in Tunis.
$$\begin{array}{ll} incidence= & ((new\;hospitalized\;cases\;of\;COVID\text{-}19\;in\;children\;aged\;0\\ & -15\;years\;(Mar\;2020\\ & -Feb\;2022))/(Population\;of\;children\;aged\;0\\ & -15\;years\;in\;Tunis))\times 100,000 \end{array}$$

The mortality rate of SARS-CoV-2 infection among the same population was calculated using the number of deaths over the study divided by the total number of infected children.
$$Infection\;fatality\;ratio\;(\%)=(\frac{Number\;of\;deaths\;from\;disease}{Number\;of\;infected\;individuals})\times 100$$

### 2.6. Ethical Considerations

The Ethics Committee of Béchir Hamza Children’s Hospital, Tunis approved the study. The study database did not contain any personal identifiers. The study adhered to the outlined protocols and abided by legal and regulatory mandates. The principles laid down by the International Ethical Guidelines for Biomedical Research Involving Human Subjects (Council for International Organizations of Medical Sciences 2002), the ICH Guideline for Good Clinical Practice, and the Declaration of Helsinki were followed.

## 3. Results

### 3.1. Baseline Characteristics

Out of 2790 hospitalized patients, 327 pediatric patients with a COVID-19 diagnosis were included in the study with a mean age of 3.3 years; over 45% of the study patients were <6 months old (Figure 1). Most of the patients (*n* = 183, 57%) included in the study were male (Table 1). The distribution of confirmed cases among the different provinces revealed that the majority of the included patients (*n* = 207, 64.5%) were from Tunis Governorate.

Further, the temporal distribution of confirmed COVID-19 cases years showed a rise in the number of pediatric cases in June 2021 (Figure 2), reflecting the time of surge in the overall number of cases within Tunisia [4].

A little more than a quarter (26.5%; *n* = 85) of the hospitalized COVID-19 children had at least one comorbidity; neurological disease was the most common comorbidity (20%: 17/85) reported among the study patients, followed by asthma and congenital disease (both 16.5%: 14/85). Fever was the most common primary reason for hospitalization (49%), followed by respiratory problems (32%). Table 1 presents the baseline characteristics, underlying comorbidities, and reasons for hospitalization.

Circulating variants:

Among the investigated sequences, the detected variants and lineages involved three Variants of Concern (VOCs), Alpha (*n* = 23), Delta (*n* = 60), and Omicron (*n* = 31), as well as three other lineages: B.1.160 (*n* = 7), B.1.177 (*n* = 1), and C.36.3 (*n* = 1). The delta variant was the predominant VOC among the studied population (48.8%), followed by the Omicron and Alpha variants (26% and 19.5%, respectively). The most frequent sub-variant into the Delta variant was AY.122 (98%). For the Omicron variant, the most frequent sub-variant was BA.1 (65%) followed by BA.2 (35%).

Two peaks of case numbers appeared, a Delta peak between June and August 2021 and an Omicron peak in January and February 2022 (Figure 3).

### 3.2. Study Outcomes

#### 3.2.1. Clinical Presentation

A total of 99% of the study population had symptoms of COVID-19 infection. The majority of hospitalized children presented with fever (95.28%) followed by cough (43.71%) and dyspnea (39.62). The other common clinical presentation noted among the hospitalized children is provided in Figure 4. The viral load of SARS-CoV-2 was moderate; the mean CT value was 26.24 (range: 13–38) for all RT-PCR samples in the study population (SARS-CoV-2 viral load is considered moderate for CT values ranging between 24 to 35) [18].

#### 3.2.2. Disease Severity

About 30% of the study population required oxygen support while about 13% required ICU admission and 12.2% required assisted ventilation. Of the 43 children admitted in the ICU, 33 children were aged below 1 year with seven out of eight deaths occurring in this age group (Table 2). Furthermore, only 3.12% of the patients (*n* = 10) were hospitalized for more than one month, while 70% of the study population (*n* = 225) was discharged by the 5th day of hospitalization. A total of 17 deaths (5.3%) due to COVID-19 infection were reported in the study population. Eight deaths occurred in the Pediatric ICU (PICU); the others were not transferred because of the lack of ICU beds and were managed in the pediatric departments.

#### 3.2.3. Description of the Deceased Population

In this study, a total of 17 children died due to COVID-19, comprising ten boys and seven girls, with a mean age of 1.9 years. Twelve of them were aged under one year. Most of the deceased patients were hospitalized for fever, dyspnea, and cough, and the severity of infection in the deceased population was determined by the need for oxygenation (*n* = 14), ICU admission (*n* = 8), and assisted ventilation (*n* = 10). Eight of the deceased patients had an underlying comorbidity. Neurological disease and congenital heart disease were the common comorbidities reported in this population. The mean CT of viral load for the deceased was 27.6, ranging from a minimum of 16 to a maximum of 37.

#### 3.2.4. Incidence Rate of Hospitalization and Inpatient Case Fatality Rate

The incidence rate of hospitalization of COVID-19 was found to be 77.02 per 100,000. The total population of children aged ≤ 15 years in Tunis is 268,754. The inpatient case fatality rate of COVID-19 in the study population was 5%.

## 4. Discussion

The current study reports the demographic characteristics and disease outcomes of the pediatric population with COVID-19 hospitalized at Béchir Hamza Children’s Hospital throughout the various stages of the pandemic in Tunisia. This study had the largest research sample of children with COVID-19 in Tunisia. The current study reported that the incidence rate of hospitalization due to COVID-19 was 77.02 per 100,000 of the child and adolescent population (≤15 years) in Tunis during March 2020 and February 2022. So far, only one previous study assessed the hospitalization rate among Tunisian children and adolescents (≤15 years) and reported that 8.5% of all hospitalizations occurred due to COVID-19 during a similar time period [19]. In the US, the weekly hospitalization rates during 1 March 2020–19 February 2022 in children aged 0–4 years was 14.5, with highest hospitalization rates among infants aged <6 months [20]. In a separate analysis, out of 8121 patients affected with COVID-19 in the age range of 0–9 years, 5.7% required hospitalization, with in-hospital death in 0.2% patients [21]. However, the outcomes of these studies cannot be directly compared to the present study because of possible variations in statistical methods used to calculate the incidence, different demography and age group of patients, and different time periods covered under the analysis. Further, the inpatient case fatality rate in our study was 5% and this was higher than that reported in other similar studies [22,23]. The inpatient fatality rate as per the previous study conducted in Tunisia was 0.46% [19]. Similarly, the inpatient fatality rate was found to be in the range of 1.4–1.8% in the US in pediatric patients hospitalized with COVID-19 [24,25]. This high rate may be attributed to the lack of ICU beds and the young age of patients included in the study (45% of the study patients were below 6 months of age). About half of the deceased patients had underlying comorbidities, thereby emphasizing the importance of monitoring and taking control measures in this high-risk population. 

Among the 2790 hospitalized children with acute respiratory infections, the study included 327 SARS-CoV-2-positive children, with a positivity rate of 11.4%. Our study exhibited a higher positivity rate than that reported in the initial phases of the pandemic, where the positivity rate was around 2% in children [26,27]. Nevertheless, a marked increase in the number of children contracting COVID-19 infection was witnessed over time, largely attributable to changing COVID-19 associated symptoms, laboratory testing capacity, and the emergence of different variants [28,29].

The demographic characteristics in the current study demonstrated that COVID-19 infection was more frequent in children below 6 months of age and in boys compared to female children. In the present study, more than half of the patients (53%) were <1 year old, with 45% being less than 6 months old. Similar results have been reported in the meta-analysis study by Bhuiyan et al., wherein 53% of the study population was reported to be less than one year of age [30]. Furthermore, the registry study by Sobolewska-Pilarczyk et al. reported that 23% of children with COVID-19 were aged < 1 year [31]. Although a lower frequency of COVID-19 infection is evident in children, they have a higher susceptibility to contracting it during the first few months of their lives due to their inadequately developed immune systems [30]. Moreover, the majority of the deaths (12 out of a total of 17 deaths) reported, in our study, occurred to infants or children less than 1 year of age. This observation is consistent with a previous study that reported high mortality in children less than 1 year of age [32]. 

Studies have reported that among infants, although COVID-19 is symptomatically mild, even the low risk of severe infection would require prevention. The presence of comorbid conditions can heighten the risk of infections and their complications in infants [33]. Therefore, it is imperative to focus on protecting the infants from COVID-19 infections. As infants under 6 months of age are ineligible for any existing COVID-19 vaccines, maternal vaccination is the only available option in preventing disease in early infancy. It is noted that infants acquire immunity via the placenta from their mothers [33,34]. A recent study conducted in the US reported that the incidence of hospitalization for COVID-19 was considerably lower for infants aged < 6 months if their mothers were vaccinated compared with infants of unvaccinated mothers (21/100,000 person-years vs. 100/100,000 person-years) [34]. Additionally, it has been noted that infants and children are often infected by coming into contact with any COVID-19 positive patient (whether a parent or any family member). Studies have found that 70–90% of COVID-19 infected children are infected by a household contact/ family member [14,35,36]. 

Multiple studies have reported higher risk of COVID-19 in males over the period of time [10,27,37]; similarly, a male predominance was observed in our study. The modulation of transmembrane serine protease 2 expression by sex steroids has been put forward as a possible reason for male predominance noted among the COVID-19 population. The existence of TMPRSS2:ERG in prostate cancer, along with the significant modulation of TMPRSS2 by androgens, has led to the speculation that the male predominance in the COVID-19 pandemic may be partly attributed to TMPRSS2 [38].

The temporal distribution of confirmed COVID-19 cases showed a rise in the number of pediatric cases in June 2021. Tunisia experienced the fourth wave of COVID-19 with the Delta variant, which was prevalent during May 2021 [16]. Globally rapid transmission of the Omicron variant caused an unprecedented surge in COVID-19 cases and hospitalizations among children [39].

In the current study, 26% of the children had comorbidities, with neurological disease being the most common comorbidity, followed by asthma, congenital disease, and obesity. Similar to adult patients, children with comorbidities are at increased risk of severe and critical illness and mortality during COVID-19 infection. Evidence from the literature suggests the association of ≥1 comorbidities with severe COVID-19 infection among both adults and the pediatric population [40,41]. The Netherland-based retrospective study by Biharie et al. has reported similar findings, wherein 65.5% of the patients had pre-existing comorbidities and the common comorbidities in their study included obesity (21.7%), respiratory disorders (19.6%), and neurological disorders (17.4%) [42]. Similarly, a population-based surveillance study in the US reported that 55.0% of the study population had ≥1 underlying medical conditions, with obesity, chronic lung disease, neurologic disorders, cardiovascular disease, and blood disorders being the most commonly reported ones [22]. The meta-analysis study by Tsankov et al. reported that children with comorbidities are more susceptible to severe COVID-19 manifestations and associated mortality than healthy children [43]. Obesity has also contributed for severe disease manifestations in children [42,43]. In our study, obesity was reported in 10.59% of the patients. Obesity increases the risk and severity of COVID-19 and leads to nutritional, cardiac, respiratory, and immune response changes, which could worsen the complications from SARS-CoV-2 infection [44]. Additionally, obesity has been identified to increase the risk of hospitalization in the pediatric population. It often leads to immune dysregulation and increased susceptibility to infections [45]. A study conducted in New York using data from a tertiary care unit showed obesity as the most prevalent comorbidity in pediatric patients with COVID-19 [46]. 

From our study, it was also noticed that 20% of the patients had underlying neurological conditions. Neurological diseases often worsen the symptoms of respiratory diseases, leading to severe illness by reducing muscle tonality and impairing mobility and structural conditions, in turn reducing pulmonary function [47,48]. The COVID-19 pandemic has impacted individuals with neurological conditions, complicating their ability to obtain diagnostic tests, treatments, and therapies [49]. 

These results substantiate the need to identify and monitor the comorbidities to aid in public health strategies, such as prioritizing vaccinations for at-risk children and implementing targeted prevention measures.

The most common reasons for hospitalization in the current study included fever, followed by cough. The results obtained corroborated previous epidemiological studies, where fever, cough, rhinorrhea, respiratory distress, sore throat, vomiting, and diarrhea were the most commonly reported symptoms [27,28,29,50]. The other symptoms included headaches, anorexia, myalgia, nausea, and fatigue.

In our study, nearly 30% of the patients required oxygen support while approximately 13% required ICU admission. The study by Woodruff et al. reported similar findings, wherein nearly one-third of the study population needed ICU admission or the use of invasive mechanical ventilation. They were also at a greater risk of contracting severe COVID-19 infection [22]. 

This study has some limitations. Firstly, our study being a retrospective design relied mainly on routinely collected data which might have led to unintentional bias during data extraction and reporting. Furthermore, our study does not have follow-up data on hospitalized children with COVID-19; therefore, the study results are only limited to acute disease burden and do not provide long-term burden of disease. Secondly, the patients were selected from a single hospital and hence the results might be difficult to generalize in a larger number of patients. Furthermore, bias in the results of hospitalization rates needs to be considered. The relationship between extrinsic factors leading to variability of hospitalization among different age groups has not been captured. The geographic restriction to only Tunis does not provide the true representation of COVID-19 in all of Tunisia. Moreover, hospitalization data for different COVID-19 variables was not available in the defined period. Information bias might exist in the study due to inconsistent recording of data, underreporting, and missing data. Additionally, the possibility of missing data of patients who are less likely to be registered for primary care could not be ruled out. 

Overall, the number of studies reporting COVID-19-related disease characteristics and outcomes in children is limited in Tunisia. Therefore, the present study contributes to the evidence indicating that children can be affected by COVID-19 and may experience severe illness requiring hospitalization. The study by Borgi et al. has previously described the clinical scenario of COVID-19 infection and hospitalization in the PICU from June 2021 to August 2021 in Tunisia [15]. The study had included only 20 patients with no comorbidities. As the number of patients was limited, the findings from our study also have important implications for public health strategies focused on preventing the spread of COVID-19 infection and mitigating its impact on vulnerable populations, including children. Additionally, comprehending the factors pertaining to the transmission of severe COVID-19 infection in children could reduce infection risk through various mitigation strategies, including the prioritization of vaccinations in children. Furthermore, larger cohort studies across different regions and countries are warranted to identify the various predictors of severe disease in children.

## 5. Conclusions

COVID-19 infection is more frequent in infants less than 6 months of age compared with older children. The study highlights that children with comorbidities were more susceptible to severe COVID-19-related clinical outcomes. Indeed, more than one-third of ICU admitted patients had at least one comorbidity. Moreover, about 1 in 10 COVID-19 hospitalized children required ICU admission. 

## Figures and Tables

**Figure 1 viruses-16-00779-f001:**
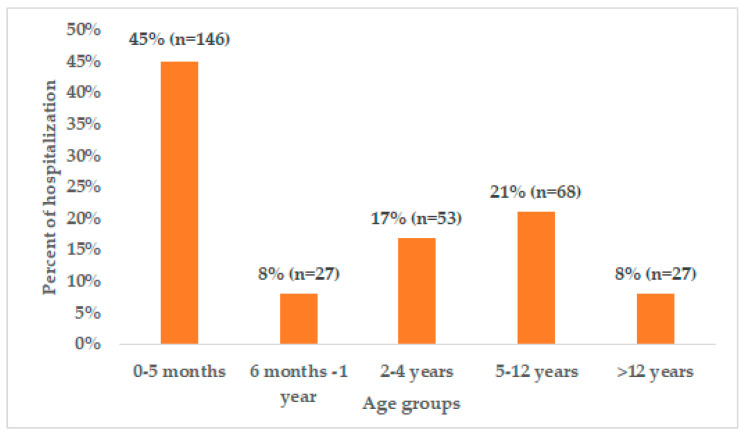
Proportion of children hospitalized due to COVID-19 infections in different age groups.

**Figure 2 viruses-16-00779-f002:**
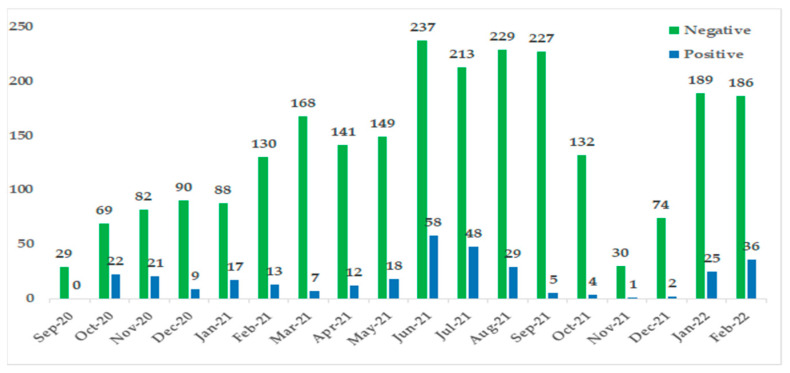
Distribution of SARS-CoV-2 test results by sample collection date.

**Figure 3 viruses-16-00779-f003:**
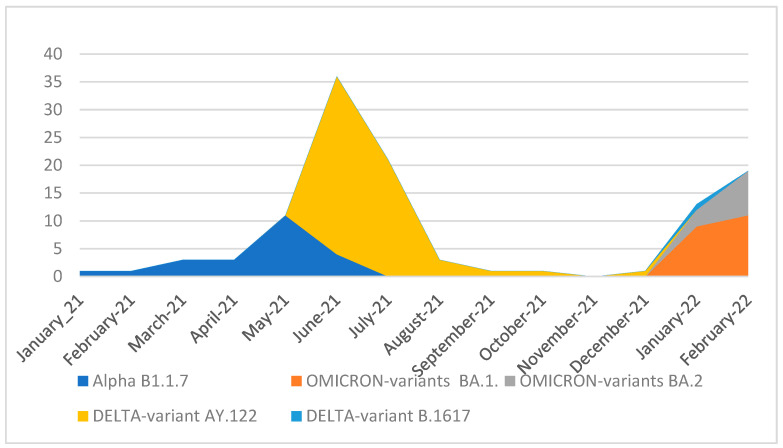
Evolution of SARS-CoV-2 circulating variants between January 2021 and February 2022.

**Figure 4 viruses-16-00779-f004:**
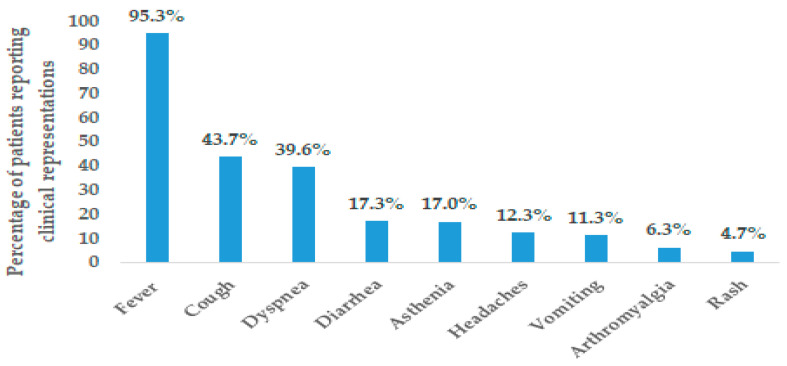
Distribution of clinical presentation of COVID-19 in children hospitalized at Béchir Hamza Children’s Hospital.

**Table 1 viruses-16-00779-t001:** Baseline characteristics of children with COVID-19 admitted to Béchir Hamza Children’s hospital.

Characteristics	All Cases (N = 327)
Age (years)	
Mean (SD)	3.3 (4.53)
Median [25, 75]	0.8 [0.14; 5.5]
Gender, *n* (%)	
Female	140 (43)
Male	187 (57)
Residence Governorate, *n* (%)	
Tunis	209 (64)
Ariana	33 (10)
Ben Arous	26 (7.9)
Mannouba	20 (6.1)
Nabeul	10 (3)
Beja	7 (2.1)
Bizerte	5 (1.5)
Siliana	4 (1.2)
Zaghouan	4 (1.2)
Jendouba	4 (1.2)
Kasserine	3 (0.9)
Elkef	1 (0.3)
Kairouan	1 (0.3)
Comorbidities *, *n* (%)	
Neurological disease	17 (20)
Asthma	14 (16.47)
Congenital heart disease	14 (16.47)
Obesity	9 (10.59)
Diabetes	7 (8.24)
Others **	47 (55.29)
Reasons for hospitalization, *n* (%)	
Fever	157 (49)
Respiratory problems	103 (32)
Seizures	17 (5)
Gastric problems	15 (5)
Cardiovascular problems	5 (2)
Glycemic problems	4 (1)
Others	29 (9)

* Patients can have more than one condition; ** Other comorbidities included genetic disease, cancer, congenital disease, autoimmune disease, and hepatic disease.

**Table 2 viruses-16-00779-t002:** Total number of children hospitalized in ICU due to COVID-19 infections.

	0–5 Months	6 Months–1 Year	2–4 Years	5–12 Years	>12 Years
Total	23	10	3	6	1
Deaths	5	2	0	1	0
Comorbidities *					
Obesity	0	0	0	5	1
Neurological disease	1	0	0	1	0
Heart disease	2	0	0	0	0
Asthma	0	0	0	1	0
Diabetes	0	0	0	0	1
Liver disease	0	1	0	0	0
Inherited metabolic disease	0	1	0	0	0

* Patients can have more than one condition.

## Data Availability

The data presented in this study are available upon request from the corresponding author.

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
