# Peer review of "Retrospective Analysis of Clinical Characteristics and Disease Outcomes in Children and Adolescents Hospitalized Due to COVID-19 Infection in Tunisia"

_viruses, 2024, doi:10.3390/v16050779_

Round 1
Reviewer 1 Report
Comments and Suggestions for Authors
General comment
The paper “Retrospective Analysis of Clinical Characteristics and Disease 2 Outcomes in Children and Adolescents Hospitalized due to 3 COVID-19 Infection in Tunisia” is an interesting article that explore an important issue: incidence, clinical characteristics, and outcomes of SARS-CoV-2 infection among children hospitalized at Béchir Hamza Children's Hospital.
Major comments
The authors should better explain that the study is based on a description of a pediatric case serie of COPVID-19 from a hospital in Tunisia. They should avoid confusion throughout the text between Tunisia and the governorate of Tunis. A map of Tunisia with the origin of the cases would help to better understand the distribution of cases.
The calculation of incidence rate of COVID-19 hospitalization in Tunis governorate is wrong (please use only numerator and denominator of cases o populations of Tunis governorate). The fatality rate should be inpatient case-fatality rate along the article.
The severity (CIU and death) should be studied by age groups (>1 year, 1-5 and 6-15) and with the presence or absence of comorbidities.
Specific comments
1) Review incidence of hospitalization and fatality rate. Point out the special severity in children under 1 year of age with obesity and neurological diseases.
2) Review the objectives of the study (Incidence rate of hospitalization and inpatient case-fatality rate ?
3) In Methods, better explain the database study population, statistical analysis (death and CUI by age group and comorbidity?)
4) The label “Incidence rate” in Figure 1 is wrong.
5) Figure 2: the number of positive is higher than 321. Check please
6) Rewrite first paragraph of Discussion and comment on Incidence rate of hospitalization and inpatient case-fatality rate.
7) Comment in greater detail on children under 1 year of age, obesity and neurological diseases.
8) Comment on other limitations related to the variability of hospitalizations, with the recording of data and missing data.
9) emphasize the importance of vaccinating family members of children under 1 year of age and with risk factors
Comments on the Quality of English Language
Minor editing of English language required
Author Response
Please see the attachment for responses to the reviewers comments

Reviewer 2 Report
Comments and Suggestions for Authors
The paper reports the analysis of clinical characteristics and disease outcomes due to COVID-19 infection in children and adolescents hospitalized at Bechir Hamza Children's Hospital in Tunisia. References provided are adequate.
Comments:
1) The study doesn't mention the causative variant of COVID-19 in children. Mention and categorize the study even further based on variants.
2) Give more details on the neurological disease comorbidity mentioned in the study.
3) Explain more on the role of TMPRSS2 on male dominance in COVID-19 occurrence. The paper cited itself recommends more studies are required in a randomized-controlled trials.
Round 2
Reviewer 1 Report
Comments and Suggestions for Authors
General comment
The authors of the article “Retrospective analysis of the clinical characteristics and outcomes of disease 2 in children and adolescents hospitalized due to COVID-19 infection 3 in Tunisia” have introduced some of the recommendations made in the review but the article could be improved if you take into account:
1) Provide a map of Tunisia with the origin of the cases to understand the distribution and origin of the cases.
2) Rewrite the first paragraph of the Discussion and comment on the hospitalization incidence rate and the inpatient fatality rate.
3) Comment in greater detail on children under 1 year of age, obesity and neurological diseases.
4) Comment on other limitations related to the variability of hospitalizations, data recording and missing data.
5) Emphasize in the Discussion the importance of vaccinating relatives of children under 1 year of age and with risk factors
Comments on the Quality of English LanguageNone
